# A Comprehensive Quality Analysis of Different Colors of Medicinal and Edible Honeysuckle

**DOI:** 10.3390/foods12163126

**Published:** 2023-08-20

**Authors:** Zhenying Liu, Yunxia Cheng, Zhimao Chao

**Affiliations:** Institute of Chinese Materia Medica, China Academy of Chinese Medical Sciences, Beijing 100700, China; liuzy9607@163.com (Z.L.); cyx2982753537@163.com (Y.C.)

**Keywords:** honeysuckle, color, quality, physicochemical properties, metabolomics

## Abstract

Honeysuckle (the dried flower bud or opening flower of *Lonicera japonica* Thunb.), a medicinal and edible substance, has is greatly popular among consumers for its remarkable health effects, such as antioxidant, antibacterial, and anti-inflammatory effects. However, due to the influences of processing methods, storage conditions, and other factors, honeysuckles show different colors which can directly reflect the quality and the price on the market. In order to comprehensively compare the quality of different colors, 55 batches of honeysuckle samples were collected and analyzed. Their color parameters, chlorophyll content (chl), total phenol content (TPC), total flavonoid content (TFC), antioxidant activity (AA), main active compounds, and metabolites were measured. As a result, the initial green-white (GW) samples, a kind of highest-quality honeysuckle, had the smallest *a** value, largest *h**, chl, TPC, TFC, and AA values, and highest content of chlorogenic acid and cynaroside. There was a significant difference between GW samples and a series of discolored samples. As the color darkened or lightened, the quality gradually decreased. The yellow-brown (YB) samples were of the worst quality and were no longer available for clinical and health purposes. A series of differential metabolites, such as quercetin-7-*O*-glucoside and secologanoside, could be used as important references to evaluate the quality of differently colored samples. The metabolic profile of honeysuckle provided new insights into the process of color change and laid a foundation for further honeysuckle quality control. The correlation results showed that the *a** and *h** values significantly affect the abovementioned quality indicators and the 10 main active compounds. In other words, the color difference could directly reflect the quality and clinical efficacy. Multiple regression analysis was carried out using combined *L**, *a**, and *b** values to predict the quality of honeysuckle. This is the first time the quality of different color honeysuckle samples on the post-harvest link has been systematically compared and a demonstration of medicinal and edible substances with different colors has been provided.

## 1. Introduction

Honeysuckle, the dried flower bud or opening flower of *Lonicera japonica* Thunb. (Caprifoliaceae), has been widely used in traditional Chinese medicines (TCMs) for the treatment of swelling abscesses, deep-rooted boils and sores, throat impediment, erysipelas, heat-toxin blood dysentery, common colds caused by wind-heat, and fever in warm disease [1]. In modern clinical practice, honeysuckle has several medicinal benefits, such as the treatment of arthritis, diabetes mellitus, fever, infections, sores, and swelling [2,3]. Pharmacological studies have shown that extracts of honeysuckle have a broad spectrum of biological activity, including antioxidant [4,5], antibacterial [6], anti-inflammatory [7], antiphotoaging [8], antinociceptive [9], antiangiogenic [10], antipyretic [11], antiviral [12,13], and hepatoprotective effects [14]. A number of chemical components with diverse structures, such as iridoids, flavonoids, saponins, polyphenols, and cerebrosides, have been isolated from this TCM [15]. In addition to being medicinal, it is also commonly used in healthy food, cosmetics, wines, and tea for its specific activities and unique aroma [16]. With the transformation in medical purposes and health perspectives, people prefer to prevent disease through diet in their daily life [17]. Honeysuckle, as a medical and edible substance, has gained great popularity among consumers in the healthcare industry [18].

The color of honeysuckle is an external reflection of its quality, which is closely related to its efficacy and economic value. A previous study has shown that the greener the honeysuckle, the higher the content of phenolic acids, such as chlorogenic acid and isochlorogenic acid A [19]. Accordingly, evaluating the quality and price of honeysuckle by color is the most popular and convenient way at present, regardless of the processing method and source. For consumers, green-white honeysuckle means good quality, and other colors mean poor quality. However, the color of honeysuckle is prone to change due to the influences of processing methods, storage conditions, and other factors. It could change from initially green-white to yellow-green, yellow, yellow-white, and even yellow-brown. Different colors of honeysuckle are sold by sellers at different prices through different channels. Regarding quality evaluation, there are many reports about honeysuckle, but most of them are about its origin, variety, processing methods, components detection, and other factors [20,21]. For color, there are many reports that have focused on the different colors of honeysuckle [22,23]. However, they only studied the color and some quality indicators and did not systematically compare the quality differences in the quality indicators and chemical components of differently colored samples.

In this study, the contents of chlorophyll, total phenol, total flavonoid, and main active compounds were studied systematically combined with physiological and biochemical experiments, as well as the antioxidant activities of honeysuckle of different colors. Furthermore, the metabolites of five colors with significant biological characteristic differences were profiled, and the differential accumulated metabolites among them were studied. Moreover, the correlations between color and quality indicators and color and main active compounds were analyzed. These results further clarified the differences and provided a reference for the development and utilization of five colors of honeysuckles.

## 2. Materials and Methods

### 2.1. Materials

#### 2.1.1. Reagents

Folin–Ciocalteu phenol reagent, 1,1-diphenyl-2-picrylhydrazyl (DPPH), 6-hydroxy-2,5,7,8-tetramethylchroman-2-carboxylic acid (Trolox), ferric chloride hexahydrate, 2,4,6-tris(2-pyridyl)-strizine (TPTZ), and 2,2′-azino-bis(3-ethylbenzothiazoline-6-sulfonic acid) diammonium salt (ABTS) were purchased from Sigma-Aldrich Chemical Co. (St. Louis, MO, USA). Hydrochloric acid, potassium persulfate, acetic acid, sodium carbonate, sodium hydroxide, aluminum nitrate, and sodium nitrite were purchased from Macklin Biochemical Co., Ltd. (Shanghai, China). 

All standard compounds, including chlorogenic acid (No.PS000631), cryptochlorogenic acid (No.PS001110), neochlorogenic acid (No.PS000974), 3,5-*O*-dicaffeoylqunic acid (No.PS001052), 3,4-*O*-dicaffeoylqunic acid (No.AF8062803), 4,5-*O*-dicaffeoylqunic acid (No.PS001057), caffeic acid (No.PS010522), cynaroside (No.PS0342-0020), rutin (No.PS012206), and quercetin (No.PS010462), whose mass purities are ≥98%, were purchased from Push Bio-Technology Co., Ltd. (Chengdu, China). HPLC-grade solvents, including methanol, acetonitrile, and formic acid, were purchased from Fisher Scientific (Waltham, MA, USA). Analytical-grade methanol solvents were purchased from Fuyu Fine Chemical Co., Ltd. (Tianjin, China).

#### 2.1.2. Sample Collection

Fifty-five honeysuckle samples were collected from hospitals, markets, and processing households. The detailed information is shown in Appendix A. As shown in Figure 1, they were divided into five different colors: green-white (GW), yellow-green (YG), yellow (Y), yellow-brown (YB), and yellow-white (YW).

All samples were pulverized into powder (particle size: 0.425 mm) using a high-speed pulverizer, packed in sealed bags, and placed in a refrigerator at −4 °C.

### 2.2. Colorimetric Analysis

Colorimetric analysis was performed via tristimulus CIE color measurement system [24]. The CIE *L***a***b** formula defines color by breaking it down into three components (*L**, *a**, and *b**) in a three-dimensional measurement [25,26]. Among them, the *L** value indicates lightness, the *a** value takes a positive value for redness and a negative value for greenness, and the *b** value takes a positive value for yellowness and a negative value for blueness [27]. The chroma (*C**) and hue (*h**) values can be computed together by converting these coordinates from rectangular form to polar form. The *C** value is the radial component and the *h** value is the angular component of the polar representation [26].

The color difference value (Δ*E*) represents the ability of human eyes to distinguish different colors. The larger the value, the more significant the color difference between the measured sample and reference [28]. It is calculated by the following expression:(1)$$\Delta E=\sqrt{{\left(L^{*}-L_{0}^{*}\right)}^{2}+{\left(a^{*}-a_{0}^{*}\right)}^{2}+{\left(b^{*}-b_{0}^{*}\right)}^{2}}$$

Subsequently, the color parameters were determined using a NH-310 colorimeter (3NH Technology Co., Ltd., Shenzhen, China). The measurement conditions were as follows: measuring aperture was 8 mm, standard deviation was Δ*E ab* ≤ 0.06, light source was D65 (equivalent to natural daylight, 6504 K), measuring angle was 10° (diffuse illumination), and color measurement mode was SCE (excluding specular reflection light) coupled with a sensor of photodiode array. The reference colorimeter was calibrated using a white paper (*L**_0_ = 87.803, *a**_0_ = −0.645, *b**_0_ = −3.789, *C**_0_ = 3.843, and *h**_0_ = 260.345). Each honeysuckle sample was filled into cuvettes and measured in triplicate (*n* = 3).

### 2.3. Determination of Chlorophyll

Chlorophyll, a fat-soluble pigment found in all photosynthetic organisms, mainly includes chlorophyll a (chl_a_) and chlorophyll b (chl_b_) [29]. The determination of chl_a_ and chl_b_ of honeysuckle was carried out as described in references with minor modifications [30,31]. In short, 0.5 g of the sample and 25 mL 95% of ethanol were placed in a beaker, homogenized, and ultrasonically extracted (100 W, 40 KHz) for 30 min. The extract solution was centrifuged at 3000 rpm for 5 min at 4 °C. The supernatant was used for chlorophyll determination using a TU-1810 UV-visible spectrophotometer (Puxi General Instrument Co. Ltd., Beijing, China) at 649 nm and 665 nm. The chlorophyll content was calculated using the following expression:(2)$${chl}_{a}=13.95\times A_{665}-6.88\times A_{649}$$
(3)$${chl}_{b}=24.96\times A_{649}-7.32\times A_{665}$$
(4)$${chl}_{T}={Chl}_{a}+{Chl}_{b}$$
(5)M=C×V/m
where chl_a_ and chl_b_ refer to the content of chlorophyll a and chlorophyll b in the extracted solution. A_649_ and A_665_ refer to the absorbance at wavelength 649 nm and 665 nm. The chl_T_ refers to the total chlorophyll content. *M* refers to the pigment mass fraction (%) of honeysuckle samples. *V* represents the total volume (mL) of the extracted solution and m represents the weight (g) of the sample. Each assay was carried out in triplicate (*n* = 3).

### 2.4. Determination of Total Phenol and Total Flavonoid

The TPC of honeysuckle samples was determined according to the Folin-Ciocalteu method with some modifications [27]. In brief, 150 μL methanol extract solution was mixed with 6 mL water and 500 μL 2 M Folin-Ciocalteu phenol reagent in brown volumetric flask. Then, this was followed by incubation in the dark for 4 min. Next, 150 μL 20% sodium carbonate solution was added to the solution and incubated in the dark for 60 min. The absorbance was measured at 700 nm. The TPC was calculated from a standard curve with gallic acid concentration. Finally, the results were expressed as mg gallic acid equivalent per g (GAE mg/g).

The TFC of honeysuckle samples was determined using a NaNO_2_-Al(NO_3_)_3_-NaOH method with slight modifications [32]. In detail, 600 μL methanol extract solution was mixed with 6 mL water in brown volumetric flask. Then, 1 mL 5% sodium nitrite, 1 mL 5% aluminum nitrate, and 10 mL 1 mol/L sodium hydroxide were sequentially added at 0, 6, and 12 min. Subsequently, the above reacting mixture was incubated in the dark for 15 min. The absorbance was measured at 510 nm. The TFC was calculated from a standard curve with rutin concentration and expressed as mg rutin equivalent per g (RE mg/g). All solutions were used on the day of preparation. Each assay was carried out in triplicate (*n* = 3).

### 2.5. Determination of Antioxidant Activity with an ABTS Free Radical Scavenging Method

AA was measured based on the capacity of different components to scavenge the ABTS radical cation (ABTS^+^) compared to a standard antioxidant substance of Trolox with slight modifications [33,34]. Before the assay beginning, the ABTS^+^ stock solution was prepared by mixing 10 mL 7 mM ABTS solution and 176 μL 1.40 mM potassium persulfate solution, reacted in the dark for 16 h. The work solution was obtained by diluting with methanol until the absorbance reached 0.70 ± 0.02 at 734 nm. The sample solution (30 μL) was reacted with 3 mL ABTS^+^ work solution in the dark for 6 min at 23 °C. The absorbance was measured at 734 nm. The AA was calculated from a standard curve with Trolox concentration and expressed as μM Trolox equivalent per g (TE μM/g). All of the solution was used on the day of preparation. Each assay was carried out in triplicate (*n* = 3).

### 2.6. Determination of Antioxidant Activity with a DPPH Radical Scavenging Method

The AA was also determined based on hydrogen donating or radical scavenging ability using the stable radical DPPH with slight modifications [35]. In detail, 20 μL five-fold diluted sample solution and 2 mL DPPH solution were mixed well and made up to the volume with methanol. Subsequently, the reacted mixture was incubated in the dark for 30 min. The absorbance was measured at 517 nm. The calculation formula and result expression were the same as that of the ABTS^+^ method. All of the solution was used on the day of preparation. Each assay was carried out in triplicate (*n* = 3).

### 2.7. Determination of Antioxidant Activity with a Ferric Reduction Ability of the Plasma (FRAP) Free Radical Scavenging Method

A ferric reduction ability of the plasma (FRAP) method was carried out as described in references with minor modifications [36,37]. The FRAP work solution was freshly prepared by mixing acetate buffer (pH 3.6), 10 mM TPTZ solution (dissolved with 40 mM HCl), and 20 mM FeCl_3_·6H_2_O solution at a ratio of 10:1:1 (*V*:*V*:*V*). Subsequently, 20 μL of the sample solution was reacted with 5 mL FRAP work solution in the dark for 35 min. The absorbance was measured at 593 nm. The calculation formula and result expression were the same as that of the ABTS^+^ method. All of the solution was used on the day of preparation. Each assay was carried out in triplicate (*n* = 3).

### 2.8. HPLC Analysis

#### 2.8.1. Sample Preparation

The sample was prepared with minor modifications to the method described by Zheng et al. [38]. In brief, 1.0 g of the sample was extracted with 25 mL 50% methanol by means of sonication (100 W, 40 kHz) at room temperature for 30 min. The extract solution was centrifuged at 5000 rpm for 5 min at 4 °C. Additionally, the supernatant was filtered through a 0.22 μm filter and transferred to a vial. All samples were stored in a refrigerator at −4 °C until analysis. Each sample was analyzed in triplicate (*n* = 3).

#### 2.8.2. Mixed Standard Solution Preparation 

An individual standard solution of ten compounds (neochlorogenic acid, chlorogenic acid, cryptochlorogenic acid, rutin, luteolin, isochlorogenic acid B, isochlorogenic acid A, isochlorogenic acid C, caffeic acid, and quercetin) was prepared with 50% methanol, respectively. The appropriate volume of each standard solution was added to a 10 mL volumetric flask and diluted with 50% methanol to obtain the mixed stock standard solution. A working standard solution for calibration curves was prepared using a serial dilution method. All of the solution was stored in a refrigerator at −4 °C.

#### 2.8.3. Spectrometric Conditions

Chromatographic determinations were performed using a HPLC system equipped with a LC-20AT quaternary gradient pump, DGU-20A5 degasser, SIL-20A autosampler, CTD-10ASvp thermostatted column compartment, CBM-20A communication bus module, and SPD-M20A detector (Shimadzu, Kyoto, Japan). Chromatographic separation was performed using an Agilent 5 TC-C18(2) column (250 mm × 4.6 mm, 5.0 μm, Santa Clara, CA, USA). The mobile phase components were 0.1% formic acid (A) and acetonitrile (B) at a flow rate of 1.0 mL/min. The following gradient elution program was used: 0~13 min, 89~85% A; 13~25 min, 85~75% A; 25~27 min, 75~73% A; 27~45 min, 73% A. The column temperature was set at 32 °C and the injection volume was 10 μL. Additionally, the chromatographic data at a wavelength of 325 nm were applied.

#### 2.8.4. Method Validation

The HPLC method was validated according to linearity, precision, accuracy, repeatability, limit of detection (LOD), and limit of quantification (LOQ). The linearity was studied by injecting standard mixture solutions at seven concentrations and performed by plotting the peak areas versus concentration. The LOD and LOQ were obtained by injecting serial dilutions of the corresponding standard solutions, and taking the signal-to-noise (S/N) ratio of 3 and 10 as criteria, respectively. The intra-day precision was validated by injecting the standard mixture solution consecutive six times a day. The inter-day precision was validated once a day on six consecutive days. The accuracy was validated by injecting the solution at 0, 2, 4, 8, 12, 16, 24 h. The repeatability was evaluated via parallel preparation of six samples under the same condition. Precision, accuracy, and repeatability were expressed as relative standard deviation (RSD%). For the estimation of relative recoveries (R%), the found and added concentrations of the examined analytes were calculated (mean concentration found/concentration× 100, R%).

### 2.9. UPLC-Q-TOF-MS Analysis

#### 2.9.1. Sample Preparation

The sample preparation of UPLC-Q-TOF-MS was similar to that of the HPLC method, except that the extraction solvent was changed to 75% methanol, aiming to extract more chemical compositions.

Quality control (QC) samples were prepared by mixing an equal amount of each honeysuckle sample in 75% methanol, respectively. They were regarded as technical replicates to supervise the repeatability and reliability of the analytical system and were analyzed at every six injections.

#### 2.9.2. Spectrometric Conditions

The chromatographic analysis was performed on a UPLC-Q-TOF-MS system consisting of a Waters Acquity I-class UPLC and Xevo G2-XS Q-TOF mass spectrometer (Waters, Milford, MA, USA) equipped with an electrospray ionization (ESI) source. Chromatographic separation was achieved using a Waters Acquity UPLC HSS T3 (100 mm × 2.1 mm, 1.8 µm). The mobile phase consisted of 0.1% formic acid (A) and acetonitrile (B). The gradient elution was performed as follows: 0~10 min, 95~75% A; 10~15 min, 75~5% A; 15~20 min, 5% A; 20~25 min, 5~95% A. The flow rate was 0.3 mL/min, the column temperature was 45 °C, and the injection volume was 1 µL.

The MS was operated in negative ionization mode across a scan range of *m*/*z* 50 to 1200 with a scan time of 0.2 s. The following source parameters were used: capillary voltage, 3.0 kV; sampling cone voltage, 40 V; source temperature, 120 °C; cone gas, 50 L/h; desolvation temperature, 450 °C; and desolvation gas flow, 800 L/h. Argon (99.95%) was used for collision-induced dissociation and N_2_ was used as the drift gas. The low collision energy was set to 6 eV and the high collision energy ranged from 15 to 45 eV. The equipment was controlled using Masslynx 4.1 (Waters, Milford, MA, USA) software.

#### 2.9.3. Spectroscopy Processing 

The total ion chromatography was obtained and imported to the Progenesis QI (v 3.0) software (Nonlinear Dynamics, Newcastle, UK). Then, it was processed in successive steps as normalization, peak alignment, peak picking, experiment design setup, deconvolution, and metabolites identification. Subsequently, the algorithm of ANOVA *p* value and max fold change were used to filter. The specific filtration was as described by Wu et al. [39]. As for the identification of compounds, it was carried out based on their mass spectral data using the Metlin database, relevant published literature, and mixed standard solution based on their retention times and fragmentation patterns.

### 2.10. Statistical Analysis

The raw data were processed using relevant software that was compatible with the instrument and preliminarily sorted via Excel 2019 software (Microsoft, Redmond, WA, USA). All results were presented as mean ± standard deviation (SD) and analyzed via one-way analysis of variance (ANOVA) using IBM SPSS 26.0 software (SPSS, Chicago, IL, USA). Significant differences were compared using Duncan’s multiple range test. Heat-map generation was performed using MetaboAnalyst 5.0 website (https://www.metaboanalyst.ca/MetaboAnalyst/ModuleView.xhtml (accessed on 3 September 2022)). A correlation analysis was performed using IBM SPSS 26.0 software. All results were visualized via OriginLab Origin 2021 (Origin, San Francisco, CA, USA) and Graphpad Prism 9.0.0 (Graphpad, San Diego, CA, USA).

Meanwhile, these processed UPLC-Q-TOF-MS data were fed into SIMCA 14.1 software (Umetrics, Malmö, Sweden) for multivariate statistical analysis [40]. The whole data matrix was submitted to principal component analysis (PCA) and orthogonal partial least squares–discriminant analysis (OPLS-DA) to clarify the difference of differently colored samples. Then, a permutation test (200 times) was applied to validate the OPLS-DA result and to avoid over-fitting. The variable importance in projection (VIP) was used to define which compounds significantly contributed to discriminate these five color honeysuckle samples.

## 3. Results

### 3.1. Colorimetric Analysis

The color parameters of honeysuckle samples were determined by using the colorimeter combined with the CIE color space system which is widely used in the food [41,42] and herbal medicine [43,44] industries. As shown in Figure 2, the measured *L**, *a**, *b**, *C**, *h**, and calculated Δ*E* values were combined to draw a heat map and analyzed with a PCA score.

There was an obvious difference between differently colored samples. In Figure 2a, the heat map suggested that the darker the shade of brown was, the larger the color value was, while the darker the shade of blue was, the smaller the color value was. Samples with the same color could be clustered together, which was consistent with the previous color classification and could be used for the next difference analysis between groups. The results of PCA, which was an unsupervised analysis method, could reflect the original state of the data. In Figure 2b, the contribution rates of the principal component 1 (PC1) and principal component 2 (PC2) were 58.40% and 32.00%, respectively. The cumulative contribution rate of PC1 and PC 2 was 90.40% and indicated that these data could reflect the overall information of honeysuckle samples with different colors. Both GW and YW samples were well distinguished from other samples, especially the YW samples were the furthest away from other samples, indicating that YW samples were special. The Y, YG, and YB samples had partial overlap, indicating that they had a slightly significant difference.

Subsequently, box-plots were drawn (Figure 3) to further compare the color parameters. The result showed that the distribution of honeysuckle samples was different in color parameters. By contrast, GW samples had the smallest *a** value, ranging from −0.72 to 1.23 and the largest *h** value ranging from 84.71 to 92.01. YB samples had the smallest *L** value, ranging from 57.59 to 61.67. YW samples had the smallest *b*^*^ value, ranging from 7.73 to 10.12, while the *C** value’s ranged from 9.378 to 10.687, and the Δ*E* value’s ranged from 19.05 to 24.71, and the largest, the *L** value’s, ranged from 77.19 to 80.79.

In details, *L** values of GW, YG, Y, and YB samples decreased in turn, which suggested that the initial samples (GW) gradually darkened in the subsequent process of discoloration and then changed into YG, Y, and YB samples. The *L** value of the YW samples was the highest, indicating that the initial GW samples gradually faded in the subsequent process of discoloration and then changed into YW samples. The *a** value of GW samples was the smallest, indicating that the green color was the dominant hue and consistent with the representation of these samples observed by human eyes. Because the Δ*E* value has to be higher than 3.30, the color change could be observed by human eyes [45]. In this study, all Δ*E* values were significantly higher than 3.30; thus, the color could be easily distinguished in honeysuckle samples. The Δ*E* value of GW, YG, Y, and YB samples was successively increased and suggested that the color difference became increasingly significant.

### 3.2. Chlorophyll Content Analysis 

The result of chl_a_, chl_b_, chl_T_, and *M* of honeysuckle samples was shown in Figure 4. As can be seen, differently colored samples had different chl_a_, chl_b_, chl_T_, and *M* values. The distribution trend in the four chlorophyll indicators was the same, with a sequence of GW > YB > YG > Y > YW.

The GW samples had the highest chlorophyll content; chl_a_ ranged from 0.64 to 1.41 mg/mL, chl_b_ from 0.22 to 0.40 mg/mL, chl_T_ from 0.86 to 1.81 mg/mL, and *M* from 43.00 to 90.57%. Notably, the content of chl_a_ and chl_b_ in the initial GW samples showed a quantitative relationship with an approximate ratio of 3:1, which was consistent with a previous report [46]. The YW samples had the lowest chlorophyll content; chl_a_ ranged from 0.13 to 0.31 mg/mL, chl_b_ from 0.04 to 0.07 mg/mL, chl_T_ from 0.19 to 0.35 mg/mL, and *M* from 9.85 to 17.70%. It was found that the chl_a_ and chl_b_ content no longer followed the quantitative relationship of discolored YG, Y, YB, and YW samples. These results suggested that the chlorophyll was degraded, while the pigment ratio was also destroyed during the color change process of honeysuckles.

### 3.3. Total Phenol Content Analysis

A standard curve was drawn with gallic acid concentration and absorbance as horizontal (*X*) and vertical (*Y*) coordinates, respectively. The linear equation was *Y* = 98.5860*X* + 0.0127, the correlation coefficient *R*^2^ was 0.9996, and the linear range was 1.92~7.69 μg/mL. The TPC of honeysuckle samples was calculated via a standard curve and shown in Figure 4.

The GW samples had the highest value, ranging from 20.34 to 235.24 GAE mg/g, and the YB samples had the lowest value, ranging from 11.34 to 17.81 GAE mg/g.

There were significant differences among differently colored samples. The TPC values in all discolored samples of YG, Y, YB, and YW were lower than those in the initial GW samples. The results indicated that the content of phenolic compounds decreased when the color of honeysuckles changed.

### 3.4. Total Flavonoid Content Analysis

A standard curve was drawn with rutin concentration and absorbance as horizontal (*X*) and vertical (*Y*) coordinates, respectively. The linear equation was *Y* = 11.4130*X* + 0.0057, the correlation coefficient *R*^2^ was 0.9997, and the linear range was 16.05~60.19 μg/mL. The TFC of honeysuckle samples was calculated via a standard curve and shown in Figure 4.

It was found that there was considerable variation in TFC in differently colored samples. The GW samples had the highest value, ranging from 55.04 to 70.78 RE mg/g, and the YB samples had the lowest value, ranging from 22.41 to 27.66 RE mg/g. Different from TPC, the TFC in YW samples was higher than those in the YG, Y, and YB samples. There was little change in flavonoid compounds in the process of changing from GW to YW. As another main component of honeysuckle, the content of total flavonoid is also an important indicator to evaluate its quality.

### 3.5. Antioxidant Activity Analysis

The standard curves were performed with Trolox concentration and free radical scavenging rate as horizontal (*X*) and vertical (*Y*) coordinates for the three methods used in the determination of AA. For the ABTS method, the linear equation was *Y* = 163.6*X* − 0.001, the correlation coefficient *R*^2^ was 0.9999, and the linear range was 0.16~4.42 μM/g. For the DPPH method, the linear equation was *Y* = 139.89*X* + 0.0294, the *R*^2^ was 0.9998, and the linear range was 0.81~4.86 μM/g. For the FRAP method, the linear equation was *Y* = 0.6028*X* + 0.3037, the *R*^2^ was 0.9996, and the linear range was 113.55~6115.54 μM/g. The AA values of all samples using the three assays were calculated and shown in Figure 4.

As can be seen in Figure 4, there were obvious variations in the AA of the differently colored samples. The results of the abovementioned three evaluation methods were generally consistent. The GW samples had the strongest AA, and the YB samples had the weakest AA. For the ABTS method, the capacity to scavenge ABTS’ radical cation of the GW samples was as high as 145.46 TE µM/g, and that of the YB samples was as low as 58.31 TE µM/g. For the DPPH method, the ability to scavenge hydrogen donating or radical cation of the GW samples was as high as 177.95 TE µM/g, and that of the YB samples was as low as 55.53 TE µM/g. For the FRAP method, the ability of ferric reduction of the GW samples was as high as 26,597 TE µM/g, and that of the YB samples was as low as 9590 TE µM/g. These results showed that the AA of honeysuckle gradually decreases with the deepening of color. Although the AA was lower for the YW samples than compared to the GW samples, the decrease was weak. It further suggested that the YW samples might not suffer from the same conditions as those of the YG, Y, and YB samples.

### 3.6. HPLC Analysis 

#### 3.6.1. Method Validation

The analytical parameters of the developed HPLC method for the determination of these compounds are summarized in Appendix A. The relative recoveries are shown in Appendix A.

As a result, ten analytes demonstrated the method had good linearity (*R*^2^ ≥ 0.9993) in a wide concentration range. The method showed good precision, demonstrated by the RSD (%) of the intra- and inter-day studies, ranging from 1.26 to 2.59% and from 1.31 to 2.91%, respectively. The accuracy and repeatability were also acceptable, ranging from 1.23 to 3.82% and from 0.97 to 3.25%, respectively. The relative recoveries ranged from 99.46 to 103.42%, indicating the good accuracy of this method.

#### 3.6.2. Samples Analysis

The validated HPLC analytical method was applied in the analysis of the honeysuckle samples. Fifty-five samples were analyzed and ten main active compounds were determined. The characteristic chromatograms are shown in Figure 5. The content distribution of these compounds of differently colored samples is shown in Figure 6.

In Figure 6, there were significant differences among the ten main active compounds of different color honeysuckle samples. The GW samples had the highest content of neochlorogenic acid (0.26%), chlorogenic acid (2.77%), cryptochlorogenic acid (0.15%), rutin (0.06%), cymaroside (0.07%), isochlorogenic acid B (0.09%), isochlorogenic acid A (2.15%), and isochlorogenic acid C (0.35%) and the lowest content of caffeic acid (0.04%) and quercetin (0.26%). This quantification result was in accordance with those of previous reports [47,48]. The Y samples had the lowest content of cymaroside (0.04%). The YB samples had the highest content of caffeic acid (0.50%) and quercetin (0.76%), and the lowest content of chlorogenic acid (0.40%), cryptochlorogenic acid (0.05%), rutin (0.01%), isochlorogenic acid B (0.04%), isochlorogenic acid A (1.08%), and isochlorogenic acid C (0.09%). This suggested that the YB samples might have the worst quality. The YW samples had the lowest content of neochlorogenic acid (0.15%). The differences of these main active compounds indicated that the chemical composition of honeysuckle samples had changed significantly during the process of discoloration.

### 3.7. UPLC-Q-TOF-MS Analysis

#### 3.7.1. Data Quality Assessment

To guarantee the repeatability and reliability of the data, the overlapping display and analysis of the mass spectrometry results of the QC samples are shown in Appendix A. The result show the high overlap ratio of total ion current (TIC) curves of QC samples, indicating that the test results were reliable.

#### 3.7.2. Overview of the Metabolites 

A total of 93 metabolites were identified and divided into 11 classes, including 24 flavonoids, 21 iridoids, 20 phenolic acids, 8 organic acids, 4 amino acids, 3 aldehydes, 3 nucleosides, 3 saponins, 2 alcohols, 2 esters, 1 alkane, and 2 others (Figure 7a). The detailed information of the metabolites is shown in Table 1.

The PCA score of all metabolites is shown in Appendix A. The results show that the honeysuckle samples with different colors were not separated except for the YB samples, which indicated that the overall metabolic differences were not significant.

#### 3.7.3. Identification of Differential Metabolites 

Combined with the process of honeysuckle color change, a pair comparison between the two groups was attempted. For example, under the influence of certain factors, GW may change first to YG, then to Y, and finally to YB. Then, multivariate statistical analysis was carried out to explore the differences between the GW and YG, YG and Y, Y and YB, and GW and YW samples.

Take the pair comparison between GW and YG samples. PCA was performed, as shown in Figure 7b. The PCA results show that the GW and YG samples were separated well, indicating that the metabolic difference was significant. The contribution rates of the PC1 and PC2 were 40.60% and 16.40%, respectively. As an unsupervised analysis method, PCA cannot ignore within-group difference and eliminate irrelevant random errors. Therefore, supervised OPLS-DA was used to further explore the differences in metabolites (Figure 7c). However, over-fitting was easy to occur while expanding the differences; therefore, it was necessary to arrange an experiment with the help of an external model validation method (*n* = 200) to prove the validity of the model. Its results indicated that the model was effective, stable, predictable, and could be used to continue to screen the differential metabolites. From the OPLS-DA scatter plot of GW and YG samples (*R*^2^X: 0.893, *R*^2^Y: 0.851, and Q^2^: 0.934), it can be seen that the two groups were clearly distinguished. Then, ANOVA |*p*| ≥ 0.05 (Figure 7d) and VIP ≥ 2 (Figure 7e) were combined to screen the differential metabolites. Next, 36 metabolites were screened and regarded as differential metabolites. Six differential metabolites (Appendix A) were finally identified by comparing the mass spectrum data with the reference standards, the literature and the data.

As shown in Figure 8, the content of secologanoside in the GW samples was significantly lower than that in the YG samples, while the contents of lamalbide, secologanic acid, 1-caffeoylquinic acid, 4-*O*-caffeoylquinic acid and quercetin 7-*O*-glucoside were significantly higher than that in the YG samples.

Similarly, multivariate statistical analysis showed that differently colored samples of YG and Y, Y and YB, and GW and YW were also well separated, indicating that the metabolic differences were significant. Next, three differential metabolites were identified between the YG and Y samples (Appendix A), eleven differential metabolites were identified between the Y and YB samples (Appendix A), and seven differential metabolites were identified between the GW and YW samples (Appendix A).

### 3.8. Correlation Analysis

The above results show the significant difference in these indicators and main active compounds of differently colored honeysuckle samples. In order to evaluate whether color has an influence on these indicators and main active compounds, the correlation was analyzed. The results of the correlation analysis between the color parameters and the quality indicators are shown in Table 2, and those between the color parameters and the main active cymarosidnds are shown in Table 3.

As a result, there were different levels of correlation among these indicators of honeysuckle samples. The *L** value had a very significant positive correlation with TFC and AA. The *a** value had a very significant negative correlation with all indicators. The *b** value had a significant positive correlation with chl_a_, chl_T_, TPC, and AA_FRAP_. The *C** value had a very significant positive correlation with TPC. The *h** value had a very significant positive correlation with all indicators. The Δ*E* value had a very significant negative correlation with TFC and AA. To sum up, the color parameters had a significant correlation with the chl_a_, chl_b_, chl_T_, TPC, TFC, and AA indicators.

As for the main active compounds, the *L** value had a very significant positive correlation with chlorogenic acid and cymarosideide, and a very significant negative correlation with neochlorogenic acid, caffeic acid, and quercetin. The *a** value had a very significant negative correlation with chlorogenic acid and cymarosideide. The *b** and *C** values did not have a significant correlation with all compounds. The *h** value had a similar significance to that of the *a** value. The Δ*E* value had a very significant positive correlation with neochlorogenic acid and caffeic acid, and had a very significant negative correlation with chlorogenic acid and cymarosideide. It can be seen that the *a** and *h** values have the most significant correlation with the main active compounds of differently colored samples, which is consistent with the results of the correlation between the color parameters and other indicators.

Although the *h** value had a significant correlation with quality indicators, the *L**, *a**, and *b** values are commonly used to evaluate the quality of food and TCMs. Therefore, these three values were used for multiple regression analysis with quality indicators to further clarify the relationship and draw the corresponding prediction. Taking TPC as an example, a multiple linear regression was carried out based on TPC as the dependent variable and three color values as independent variables. The corresponding analysis results are shown in Table 4, Table 5 and Table 6.

In Table 4, *R*^2^ was 0.540, which is larger than 0.3, indicating that the *L**, *a**, and *b** values could explain the content distribution of total phenol. The Durbin-Watson value was 1.619, which is close to 2.0, indicating that each independent variable had mutual independence and could be used for the analysis of the regression equation. In Table 5, the significance *p* < 0.01, which indicated that the multiple linear regression equation had statistical significance. In Table 6, the VIF values of all independent variables were lower than 10, indicating that there was no multicollinearity among the variables. Similarly, other indicators (chl_a_, chl_b_, chl_T_, TFC, and AA) and three color values were also analyzed, as shown in Figure 9. These multiple linear regression equations could be used to predict the quality of honeysuckle.

## 4. Discussion

Color is an important quality indicator of foods and TCMs. In general, color analysis can be carried out via visual and instrumental methods [45]. However, the description of color using a visual method is subjective and can slightly vary among different observers. An instrumental method can ameliorate the shortcomings of a visual method [49]. In this study, the medicinal and edible honeysuckle samples were divided into five different colors via a visual method. Their color parameters were determined using an instrumental method coupled to a colorimeter, which has been shown to be very effective in this field [50]. The CIE color measurement system can be used for its ability to capture tiny color difference [51,52]. All samples were classified into five categories by following an instrumental method, which was consistent with the results of the visual method. Furthermore, the same results, obtained via both visual and instrumental methods, provided a better foundation for the subsequent detection of antioxidant and other quality indicators.

The results of both the smallest *a** value and the largest *h** value of the initial GW samples were characteristic of high-quality honeysuckle. The YW samples with the smallest *b**, *C**, and Δ*E* values and the largest *L** value can indicate that the process from initial GW to final YW was color fading. On the contrary, the YB samples with the smallest *L** value can indicate that the process from initial GW to final YB was color darkening. In this way, the color changes observed by our human eyes could be adequately expressed on the instrument.

The quality difference of differently colored foods and TCMs, such as *Malva neglecta* Wallr. [53], herbal infusions [54], and red pepper [55], was studied by combining common indicators, including chlorophyll, TPC, TFC, and AA. Our determined results of chl_a_, chl_b_, chl_T_, TPC, TFC, and AA revealed that there was a significant difference among the different colors of honeysuckle samples.

The results that the content of chl_a_, chl_b_, and chl_T_ of the initial GW samples was higher than that of the YG, Y, YB, and YW samples suggest that the process of color change from GW to those discolored samples was accompanied by the degradation of chlorophyll. Similar results have been observed in some foods [56,57]. The results of the TPC were largest in the GW samples and smallest in YB samples, which can illustrate the degradation of the phenolic compounds. The results of TFC were largest in the GW samples and smallest in the YB samples, which can illustrate the degradation of the flavonoid compounds. The strength of AA usually comes from the contribution of TPC and TFC. The results of AA were strongest in the GW samples, which can suggest that the quality of GW samples was the best. On the contrary, the quality of the discolored samples was worse.

For the ten main active compounds, the content distribution in differently colored samples was significantly different. The content of chlorogenic acid of the five color samples was, in turn, GW > YW > YG > Y > YB. All of the samples met the relevant standards, except for the YB samples [1]. The content of cymarosideide of the five color samples was consistent with that of chlorogenic acid, but only the GW, YW, and some of the YG samples were in line with the relevant standards. It is worth noting that there was no similar trend among the ten main active compounds in differently colored samples, which was the reason for the different sources, storage environments, and storage time periods.

For metabolites, there were many differential metabolites with differently colored samples, indicating that the process of the initial honeysuckle changing from GW to other colors was different. This was consistent with the result that differential metabolites were selected in the process of honeysuckle changing from green-white to yellow and then to white during its growth and development [23]. Quercetin-7-*O*-glucoside was found to be a common differential metabolite of differently colored samples. It shows strong inhibition activity against influenza A and B viruses by inhibiting viral RNA polymerase and occupying the binding site of m^7^GTP on the viral PB2 protein [58]. The content of quercetin-7-*O*-glucoside significantly decreased with the color change, which indicates that the corresponding antiviral activity significantly decreased, and reflects that the quality of honeysuckle decreased after the change of color.

These differential metabolites, mainly including phenolic acids, iridoids, and flavonoids, were the active compounds of honeysuckle [59]. They have active phenolic hydroxyl and are prone to oxidizing and polymerizing to produce some dark compounds under the influence of environmental factors and the action of related enzymes [60,61]. Subsequently, these reactions lead to the darkening or fading of honeysuckle samples.

In addition, it was shown that the *a** and *h** values were significantly correlated with all of the quality indicators and main active compounds. The *a** value represents red and green color, and its numerical difference is reflected in the appearance of honeysuckle of different colors. This correlation was consistent with the theoretical results. Although *h** is not the main color parameter, a related study has shown that it is the most reliable indicator of color degradation [62], which is consistent with the correlation results of our study. Moreover, the establishment of these multiple linear regression equations allows researchers to predict quality indicators without the use of experiments. 

## 5. Conclusions

In this study, a comprehensive quality analysis of differently colored honeysuckle samples was carried out to explore their differences. Our results show that there was a significant difference in antioxidant activity, the contents of chlorophyll, total phenol, total flavonoid, and in the ten main active compounds among the differently colored samples. The GW samples had the best quality, including the highest content of chlorophyll, total phenol, total flavonoids, and antioxidant activity and main active compounds such as chlorogenic acid. After the color change, the quality indicators of honeysuckle decreased to different degrees. The YB samples had the worst quality and cannot be used in foods and TCMs. The series of differential metabolites selected could be used as important references to evaluate the quality of differently colored samples. As for the correlation analysis, the *a** and *h** values were significantly correlated with all of the quality indicators and main chemical compounds. Furthermore, a multiple regression analysis was carried out with the combined *L**, *a**, and *b** values to predict the quality of honeysuckle without any complicated experiments.

## Figures and Tables

**Figure 1 foods-12-03126-f001:**
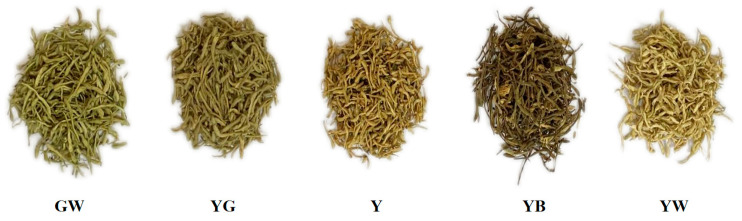
Honeysuckle samples with different colors. GW: green-white; YG: yellow-green; Y: yellow; YB: yellow-brown; YW: yellow-white.

**Figure 2 foods-12-03126-f002:**
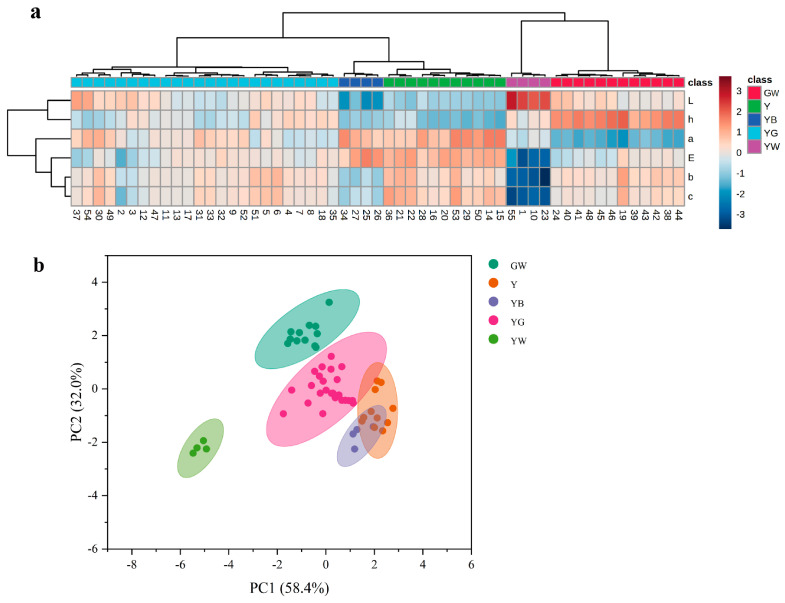
Heat map (**a**) and PCA score (**b**) of differently colored honeysuckle samples. GW: green-white; YG: yellow-green; Y: yellow; YB: yellow-brown; YW: yellow-white.

**Figure 3 foods-12-03126-f003:**
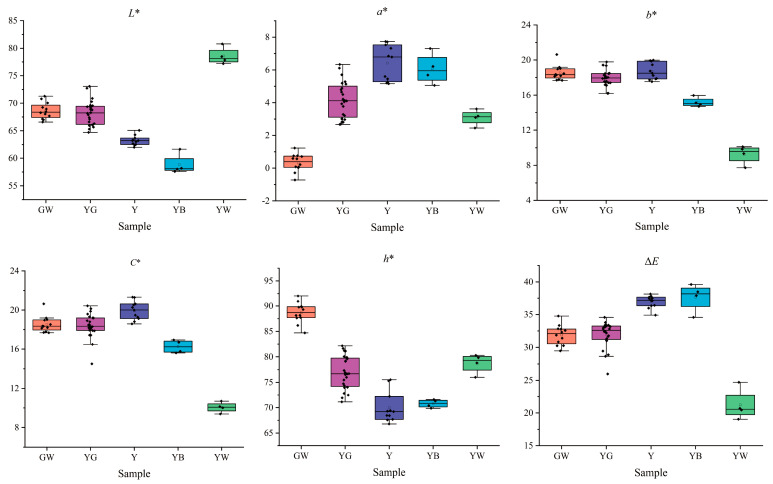
The distribution of different samples in color parameters. GW: green-white; YG: yellow-green; Y: yellow; YB: yellow-brown; YW: yellow-white.

**Figure 4 foods-12-03126-f004:**
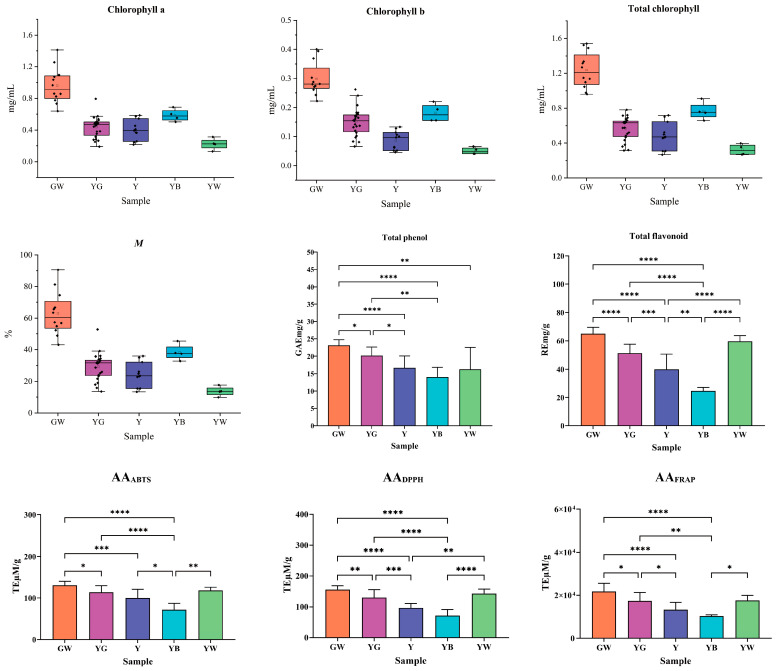
The distribution of differently colored samples in physicochemical indicators. *, *p* < 0.05; **, *p* < 0.01; ***, *p* < 0.001; ****, *p* < 0.0001. GW: green-white; YG: yellow-green; Y: yellow; YB: yellow-brown; YW: yellow-white.

**Figure 5 foods-12-03126-f005:**
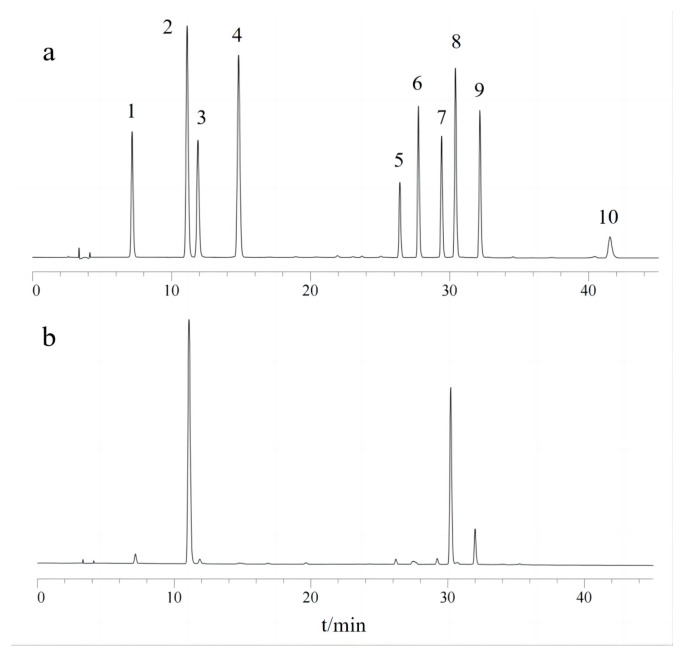
Characteristic chromatogram of mixed standard solution (**a**) and samples (**b**). (1: neochlorogenic acid, 2: chlorogenic acid, 3: cryptochlorogenic acid, 4: caffeic acid, 5: rutin, 6: cymaroside, 7: isochlorogenic acid B, 8: isochlorogenic acid A, 9: isochlorogenic acid C, 10: quercetin).

**Figure 6 foods-12-03126-f006:**
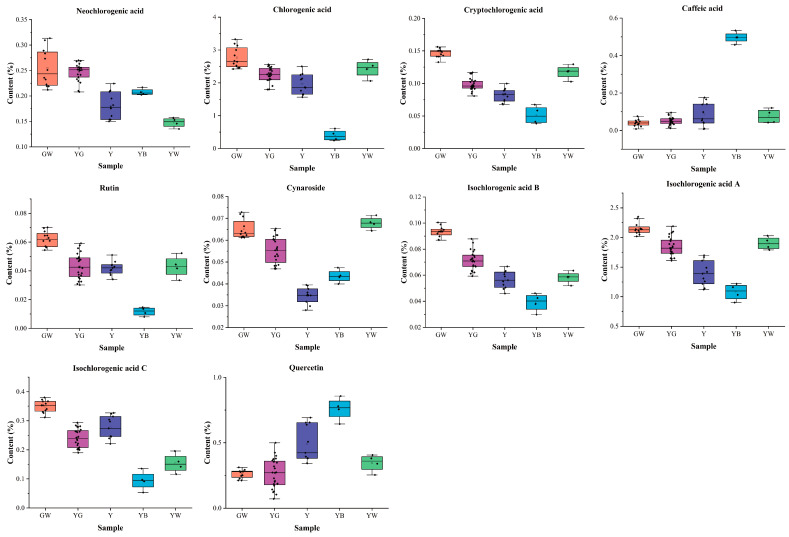
The content distribution of ten compounds of differently colored samples. GW: green-white; YG: yellow-green; Y: yellow; YB: yellow-brown; YW: yellow-white.

**Figure 7 foods-12-03126-f007:**
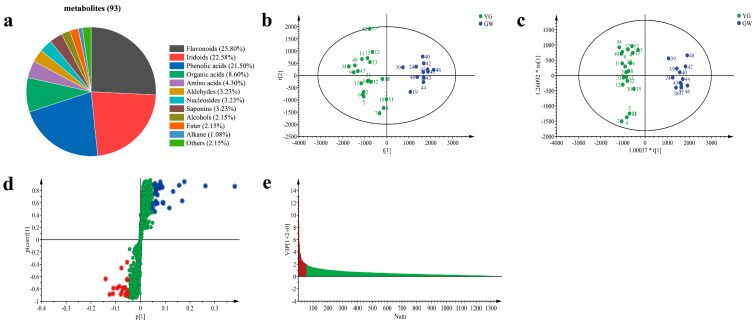
Classification of the 93 metabolites (**a**) and multivariate statistical analysis of GW and YG samples. PCA (**b**); OPLS-DA (**c**); S-plot (red and blue dots represent the value of |*p*| ≥ 0.05, green dots represent the value of |*p*| ≤ 0.05) (**d**); VIP score (red represents the value of VIP ≥ 2, gree represents the value of VIP ≤ 2) (**e**). GW: green-white; YG: yellow-green.

**Figure 8 foods-12-03126-f008:**
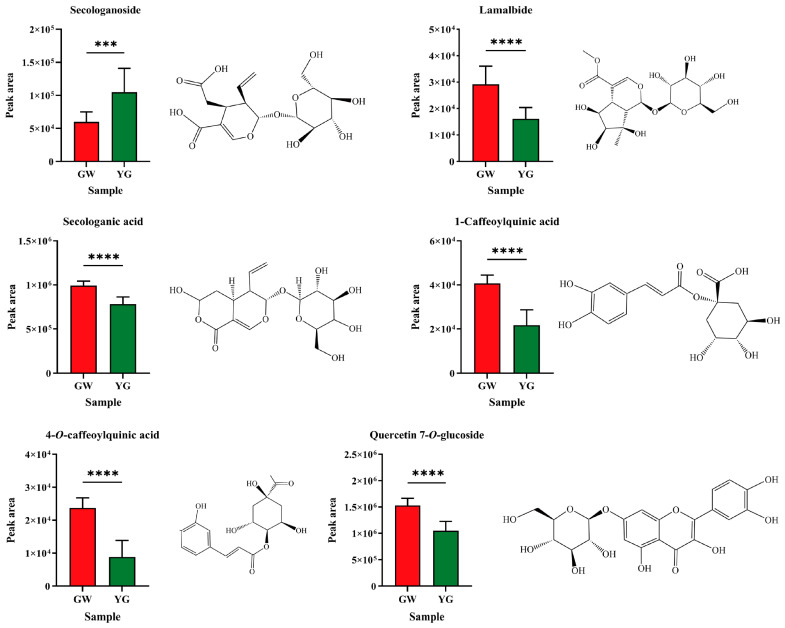
Differential metabolites in GW and YG samples. (***, *p* < 0.001; ****, *p* < 0.0001). GW: green-white; YG: yellow-green.

**Figure 9 foods-12-03126-f009:**
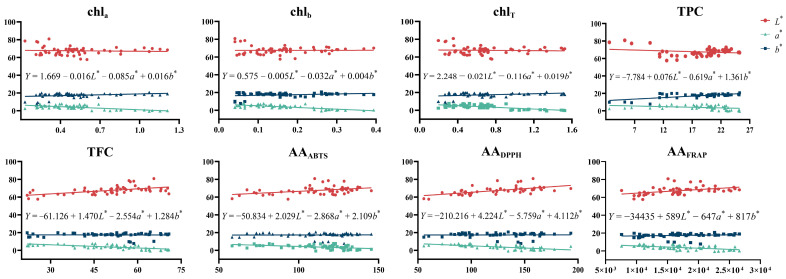
The multiple linear regression analysis of honeysuckle between color parameters and quality indicators. chl_a_: chlorophyll a; chl_b_: chlorophyll b; chl_T_: total chlorophyll; TPC: total phenol content; TFC: total flavonoid content; AA: antioxidant activity.

**Table 1 foods-12-03126-t001:** Information for the identification of the metabolites from honeysuckle samples via UPLC-Q-TOF-MS in negative mode.

No.	Identification	Rt (min)	Protonated Molecular Ion	Molecular Formula	Error (ppm)	Fragment Ions	Classification
1	Glutamine	0.77	[M − H]^−^	C_5_H_10_N_2_O_3_	−2.261	102.0551, 88.0404	Amino acid
2	5-Hydroxymethyl-2-furancarbox Aldehyde	0.79	[3M − H]^−^	C_6_H_6_O_3_	−3.837	215.0350, 126.0316	Aldehyde
3	Inosine	0.79	[M − H]^−^	C_10_H_12_N_4_O_5_	−4.551	237.0629, 195.0510, 191.0562, 179.0561	Nucleoside
4	*α*-D-Glucopyranose	0.83	[M − H]^−^	C_6_H_12_O_6_	−1.048	149.0455, 129.0189, 113.0237, 101.0236	Others
5	Malic acid	0.87	[M − H]^−^	C_4_H_6_O_5_	−5.082	133.0137, 115.0029, 71.0079	Organic acid
6	Citric acid	0.85	[M − H]^−^	C_6_H_8_O_7_	−0.075	111.0084, 87.0126	Organic acid
7	Uridine	1.06	[M − H]^−^	C_9_H_12_N_2_O_6_	−1.537	243.0615, 143.0728, 85.0286	Nucleoside
8	L-(-)-Tyrosine	1.09	[M − H]^−^	C_9_H_11_NO_3_	−2.909	180.0659, 163.0400	Amino acid
9	Succinic acid	1.18	[M − H]^−^	C_4_H_6_O_4_	−6.460	117.0186, 114.0553, 96.9622	Organic acid
10	Secologanoside	1.84	[M − H]^−^	C_16_H_22_O_11_	−0.347	389.1085, 227.0561, 209.0450, 183.0657, 165.0555	Iridoid
11	Phenylalanine	1.87	[M − H]^−^	C_9_H_11_NO_2_	−4.009	164.0709, 146.0444, 103.0527	Amino acid
12	Pantothenic acid	2.07	[M − H]^−^	C_9_H_17_NO_5_	−2.742	218.1026, 146.0811	Organic acid
13	Protocatechuic acid	2.23	[M − H]^−^	C_7_H_6_O_4_	−3.977	153.0186, 109.0285, 108.0207	Phenolic acid
14	Neochlorogenic acid *	2.49	[M − H]^−^	C_16_H_18_O_9_	−0.631	191.0560, 179.0347, 135.0446, 85.0295	Phenolic acid
15	Lamalbide	2.67	[M − H]^−^	C_17_H_26_O_12_	−0.688	240.0711, 191.0577, 179.0546	Iridoid
16	Loganic acid	2.78	[M − H]^−^	C_16_H_24_O_10_	−0.610	375.1294, 213.0764, 169.0865, 151.0756	Iridoid
17	L-Tryptophan	2.83	[M − H]^−^	C_11_H_12_N_2_O_2_	−3.046	203.0817, 186.0560, 116.0505	Amino acid
18	Methyl caffeate	3.02	[3M − H]^−^	C_10_H_10_O_4_	6.264	161.0242, 135.0443	Ester
19	8-Epi-loganic acid	3.13	[M − H]^−^	C_16_H_24_O_10_	−0.711	375.1295, 213.0766	Organic acid
20	3,4-Dihydroxybenzaldehyde	3.17	[M − H]^−^	C_7_H_6_O_3_	−3.708	136.0159, 108.0217	Aldehyde
21	Chlorogenic acid *	3.49	[M − H]^−^	C_16_H_18_O_9_	−0.645	191.0575, 127.0315, 85.0295	Phenolic acid
22	1-*O*-methyl-myo-inositol	3.58	[M − H_2_O − H]^−^	C_7_H_14_O_6_	−1.722	115.0401	Alcohol
23	Morroniside	3.59	[M − H]^−^	C_17_H_26_O_11_	−0.431	405.1402, 375.1292, 343.1034, 243.0871, 195.0657, 179.0551	Iridoid
24	Vanillic acid	3.60	[3M − H]^−^	C_8_H_8_O_4_	−4.013	375.1293, 243.0871, 195.0657, 123.0451	Phenolic acid
25	Dimethyl secologanoside	3.76	[M − H]^−^	C_18_H_26_O_11_	−2.787	353.0875, 191.0561, 173.0455, 155.0350	Iridoid
26	Quinic acid	3.77	[M − H_2_O − H]^−^	C_7_H_12_O_6_	−2.208	173.0455, 137.0239, 93.0346	Phenolic acid
27	Cryptochlorogenic acid *	3.77	[M − H]^−^	C_16_H_18_O_9_	−0.592	353.0874, 191.0561,127.0446	Phenolic acid
28	Harpagide	3.85	[M − H_2_O − H]^−^	C_15_H_24_O_10_	0.258	345.1190, 209.0454, 183.0661, 165.0555, 101.0237	Iridoid
29	Apioside	3.97	[M − H_2_O − H]^−^	C_26_H_28_O_14_	−2.645	527.1195, 353.0872, 215.0562, 97.0295	Iridoid
30	Caffeic acid *	3.99	[M − H]^−^	C_9_H_8_O_4_	2.831	135.0446	Phenolic acid
31	Methyl benzoate	4.01	[M − H]^−^	C_8_H_8_O_2_	−4.546	135.0445, 134.0367, 121.0295	Ester
32	Secologanic acid	4.16	[M − H]^−^	C_16_H_22_O_10_	0.011	193.0505, 149.0603, 105.0361	Iridoid
33	3,5-*O*-Dicaffeoylquinic ethyl ester	4.26	[M − H]^−^	C_27_H_28_O_12_	−4.231	507.1718, 357.1187, 191.1178	Phenolic acid
34	Secoxyloganin	4.30	[M − H]^−^	C_17_H_24_O_11_	−0.060	149.0504, 121.0193	Iridoid
35	1-*O*-Caffeoylquinic acid	4.63	[M − H]^−^	C_16_H_18_O_9_	−0.235	191.0561, 179.0455, 135.0295	Phenolic acid
36	4-*O*-Caffeoylquinic acid	4.64	[M − H]^−^	C_16_H_18_O_9_	−0.673	191.0561, 85.0295	Phenolic acid
37	4′-*O*-*β*-D-glucosyl-cis-p-coumaric acid	4.79	[M − H_2_O − H]^−^	C_15_H_18_O_8_	8.857	307.0824, 269.1027, 179.0561, 161.0450, 113.0245, 101.0236	Phenolic acid
38	7-*O*-ethyl sweroside	4.80	[M − H]^−^	C_18_H_26_O_10_	−1.834	401.1448, 175.0450, 101.0236	Iridoid
39	Kaempferol-3-*O*-*β*-D-rutinoside	4.92	[M − H]^−^	C_27_H_30_O_15_	−0.063	447.1185, 285.0418	Flavonoid
40	Sweroside	5.11	[M + HCOO]^−^	C_16_H_22_O_9_	−0.543	151.0660, 149.0560, 125.0239	Iridoid
41	Guanosine	5.20	[M − H]^−^	C_10_H_13_N_5_O_5_	1.451	282.0816, 133.0524	Nucleoside
42	Loganin	5.41	[M + HCOO]^−^	C_17_H_26_O_10_	−2.289	389.1721, 227.1130	Iridoid
43	3-*O*-feruloyl-D-quinic acid	5.44	[M − H]^−^	C_17_H_20_O_9_	−1.071	191.0561, 173.0451, 117.0353	Phenolic acid
44	Quercetin 3-*O*-sambubioside	6.12	[M − H]^−^	C_26_H_28_O_16_	0.605	300.0275, 271.0248, 151.0037	Flavonoid
45	7-Epi-vogeloside	6.38	[M − H]^−^	C_17_H_24_O_10_	−1.784	387.1290, 373.1139, 255.0765, 179.0546, 155.0345, 101.0237, 89.0235	Iridoid
46	Quercetin 7-*O*-glucoside *	6.38	[M − H]^−^	C_21_H_20_O_12_	0.805	301.0275, 271.0254, 151.0032	Flavonoid
47	Ferulic acid *	6.46	[M − H]^−^	C_10_H_10_O_4_	5.645	149.0621, 133.0256	Phenolic acid
48	Rutin *	6.70	[M − H]^−^	C_27_H_30_O_16_	0.805	301.0276	Flavonoid
49	L-Phenylalanino secologanin	6.70	[M − H]^−^	C_26_H_35_NO_11_	0.648	300.0276, 271.0249, 164.0714	Iridoid
50	Hyperoside	6.78	[M − H]^−^	C_21_H_20_O_12_	−0.157	301.0274, 283.0248, 255.0298, 151.0037	Flavonoid
51	Luteolin-7-*O*-neohesperidoside	7.05	[M − H]^−^	C_27_H_30_O_15_	0.380	447.0925, 285.0403, 135.0451	Flavonoid
52	Luteolin	7.23	[M − H]^−^	C_15_H_10_O_6_	−2.078	199.0428, 151.0037, 133.0281	Flavonoid
53	Cynaroside	7.24	[M − H]^−^	C_21_H_20_O_11_	−0.580	447.0929, 327.0510, 285.0404, 133.0295	Iridoid
54	Secologanin dimethyl acetal	7.41	[M − H_2_O − H]^−^	C_20_H_34_O_11_	−1.676	433.1647, 373.1133, 353.0871, 291.0866, 191.0557, 173.0451	Iridoid
55	Lonicerin *	7.46	[M − H]^−^	C_27_H_30_O_15_	0.202	285.0104	Flavonoid
56	3′,4′,5,5′,7-Pentamethoxyflavone	7.51	[2M − H]^−^	C_20_H_20_O_7_	7.741	743.2399, 729.2189, 179.0350	Flavonoid
57	Kaempferol-3-*O*-Rutinoside	7.73	[M − H]^−^	C_27_H_30_O_15_	0.270	593.1511, 285.0400, 255.0299	Flavonoid
58	Isochlorogenic acid B *	7.80	[M − H]^−^	C_25_H_24_O_12_	−2.700	353.0874, 191.0561, 179.0350, 173.0455, 161.0238, 135.0446	Phenolic acid
59	Isochlorogenic acid A *	7.98	[M − H]^−^	C_25_H_24_O_12_	−3.419	353.0877, 191.0372, 179.0348, 173.0452, 135.0445	Phenolic acid
60	Isochlorogenic acid C *	8.04	[M − H]^−^	C_25_H_24_O_12_	−2.925	353.0876, 191.0564, 179.0351, 173.0485, 155.0362, 135.0445	Phenolic acid
61	Quercetin *	8.10	[M − H]^−^	C_15_H_10_O_7_	0.431	301.0658, 193.0315, 151.2276	Flavonoid
62	Isorhamnetin-3-*O*-glucoside	8.14	[M − H]^−^	C_22_H_22_O_12_	−0.469	314.0423, 285.0399	Flavonoid
63	Kingiside	8.18	[M − H]^−^	C_17_H_24_O_11_	−1.296	165.0558, 149.0261, 119.0322	Iridoid
64	Paeonol	8.23	[3M − H]^−^	C_9_H_10_O_3_	−4.502	497.1817, 461.2023, 395.1917, 261.1337, 96.9686	Phenolic acid
65	Isomer	8.29	[M − H]^−^	C_33_H_44_O_19_	0.124	511.1104, 467.1526, 339.1299, 287.1401, 255.1931	Iridoid
66	Apigetrin	8.48	[M − H]^−^	C_21_H_20_O_10_	−0.686	431.0980, 339.0510, 268.0375	Flavonoid
67	Isorhamnetin-3-*O*-*β*-D-rutinoside	8.53	[M − H]^−^	C_28_H_32_O_16_	−0.815	315.1563, 300.0455	Flavonoid
68	Viscumneoside III	8.53	[M − H_2_O − H]^−^	C_27_H_32_O_15_	0.003	577.1563, 569.1875, 535.1458, 195.0662, 151.0764	Flavonoid
69	7-Hydroxycoumarin	8.76	[M − H]^−^	C_9_H_6_O_3_	−3.108	161.0240, 150.0332, 137.0239, 135.0447, 133.0289	Flavonoid
70	Chrysoeriol 7-*O*-glucoside	8.92	[M − H]^−^	C_22_H_22_O_11_	−0.597	298.0561, 283.0506, 255.2143	Flavonoid
71	Tricin 7-*O*-*β*-D-glucoside	8.99	[M − H]^−^	C_23_H_24_O_12_	−1.523	491.1188, 476.0960, 447.0924	Flavonoid
72	Harpagoside	9.12	[M − H]^−^	C_24_H_30_O_11_	−3.582	493.1714, 313.1081, 179.0561, 161.0455, 71.0138	Iridoid
73	Centauroside	9.12	[M − H]^−^	C_34_H_46_O_19_	0.683	725.2298, 595.2032, 525.1613, 179.0561	Iridoid
74	Nonadecane	9.12	[3M − H]^−^	C_19_H_40_	−6.636	757.2564, 595.2034, 525.1615, 493.1714	Alkane
75	(*E*)-Aldosecologanin	9.69	[M − H]^−^	C_34_H_46_O_19_	0.087	757.2556, 577.2032, 483.1713, 367.1034, 119.0561	Iridoid
76	Cyanin chloride	9.84	[M − H_2_O − H]^−^	C_27_H_31_ClO_16_	−1.603	529.1349, 367.1028, 353.0875, 191.0561	others
77	Rhoifolin	10.11	[M − H]^−^	C_27_H_30_O_14_	1.199	413.0882, 269.0451	Flavonoid
78	3-Indoleacrylic acid	10.45	[3M − H]^−^	C_11_H_9_NO_2_	−9.405	284.0923, 252.0666, 172.0767	Organic acid
79	3,4-*O*-Dicaffeoylquinic acid methyl ester	10.50	[M − H]^−^	C_26_H_26_O_12_	−0.635	367.1018, 179.0349, 161.0241, 135.0451	Phenolic acid
80	Abscisic acid	10.80	[M − H]^−^	C_15_H_20_O_4_	−2.198	245.1382, 209.1155, 152.0914	Phenolic acid
81	Kaempferol *	10.85	[M − H]^−^	C_15_H_10_O_6_	−0.425	285.0403, 215.0299, 175.0506, 151.0037, 133.0295	Flavonoid
82	3,4,5-Tricaffeoylquinic acid	11.44	[M − H]^−^	C_34_H_30_O_15_	−0.040	677.1506, 515.1193, 179.0364, 161.0242, 135.0444	Phenolic acid
83	Madreselvin A	11.51	[M − H]^−^	C_29_H_34_O_16_	−0.654	315.0432, 300.4218	Flavonoid
84	Aloinoside A	11.64	[M − H]^−^	C_27_H_32_O_13_	0.760	563.1769, 113.0244, 101.0244	Flavonoid
85	Macranthoside B	11.76	[M − H]^−^	C_53_H_86_O_22_	1.195	1073.5543, 937.5166, 749.4481	Saponin
86	Apigenin	11.89	[M − H]^−^	C_15_H_10_O_5_	−2.416	117.0185	Flavonoid
87	Macranthoidin A	11.92	[M − H]^−^	C_59_H_96_O_27_	−0.441	911.4579, 749.4481, 603.3887, 471.3681	Saponin
88	Hydnocarpin	12.71	[M − H]^−^	C_25_H_20_O_9_	−0.968	463.1025, 285.0392, 283.0240	Flavonoid
89	Decyl aldehyde	15.54	[2M − H]^−^	C_10_H_20_O	−2.384	311.2950, 281.2475, 253.2173, 199.1704, 125.0972	Aldehyde
90	Lauric acid	15.76	[2M − H]^−^	C_12_H_24_O_2_	−2.341	399.3480, 297.2800, 255.2326	Organic acid
91	Citronellol	15.79	[2M − H]^−^	C_10_H_20_O	−0.555	311.2953, 281.2850, 255.2329, 197.1911, 183.1755	Alcohol
92	Oleanic acid	15.82	[M − H]^−^	C_30_H_48_O_3_	−1.861	438.3207, 249.1672, 203.4275, 189.3225, 133.8216	Saponin
93	Tridecylic acid	16.52	[2M − H]^−^	C_13_H_26_O_2_	−1.863	427.3793, 353.3061, 255.2326	Organic acid

Note: * Compared with the standard.

**Table 2 foods-12-03126-t002:** The correlation coefficients between color parameters and quality indicators of honeysuckle samples ^a^.

	chl_a_	chl_b_	chl_T_	TPC	TFC	AA_ABTS_	AA_DPPH_	AA_FRAP_
*L**	−0.07	0.01	−0.05	−0.21	0.55 **	0.41 **	0.66 **	0.54 **
*a**	−0.65 **	−0.73 **	−0.69 **	−0.32 *	−0.66 **	−0.48 **	−0.71 **	−0.49 **
*b**	0.32 *	0.26	0.31 *	0.67 **	−0.02	0.02	0.13	0.28 *
*C**	0.22	0.14	0.21	0.62 **	−0.09	0.01	0.06	0.14
*h**	0.68 **	0.74 **	0.72 **	0.37 **	0.68 **	0.49 **	0.75 **	0.54 **
Δ*E*	0.10	−0.03	0.07	0.25	−0.53 **	−0.42 **	−0.66 **	−0.52 **

a. * *p* < 0.05; ** *p* < 0.01. chl_a_: chlorophyll a; chl_b_: chlorophyll b; chl_T_: total chlorophyll; TPC: total phenol content; TFC: total flavonoid content; AA: antioxidant activity.

**Table 3 foods-12-03126-t003:** The correlation coefficients between color parameters and main active compounds of honeysuckle samples ^a^.

	1	2	3	4	5	6	7	8	9	10
*L**	−0.37 **	0.46 *	0.17	−0.49 **	0.09	0.65 **	0.13	0.13	−0.05	−0.32 *
*a**	0.69 **	−0.64 **	−0.06	0.41 **	0.34 *	−0.57 **	0.32 *	0.56 **	0.54 **	0.47 **
*b**	−0.06	0.18	−0.03	−0.24	0.09	−0.10	−0.07	−0.05	−0.03	−0.15
*C**	0.14	0.09	−0.02	−0.17	0.13	−0.23	−0.01	0.04	0.07	−0.08
*h**	−0.68 **	0.65 **	0.07	−0.38 **	−0.35 **	0.56 **	−0.29 *	−0.55 **	−0.51 **	−0.44 **
Δ*E*	0.42 **	−0.35 **	−0.12	0.31 *	−0.04	−0.58 **	−0.15	−0.03	0.04	0.17

a. * *p* < 0.05; ** *p* < 0.01. 1–10 represent main active compounds, which was consistent with the results shown in Figure 5.

**Table 4 foods-12-03126-t004:** Model summary ^a^.

*R*	*R* ^2^	Adjusted **R**^2^	Durbin-Watson
0.735b	0.540	0.513	1.619

a. Dependent variable: TPC. Predictors: (constant), *L**, *a**, and *b**.

**Table 5 foods-12-03126-t005:** ANOVA ^a^.

Model	Sum of Squares	Df	Mean Square	F	Sig.
Regression	752.348	3	250.783	19.991	0.000
Residual	639.796	51	12.545		
Total	1392.143	54			

a. Dependent variable: TPC.

**Table 6 foods-12-03126-t006:** Coefficients ^a^.

Model	Unstandardized Coefficients	Collinearity Statistics	
	B	Tolerance	VIF
(Constant)	−7.784		
*L**	0.076	0.535	1.868
*a**	−0.619	0.770	1.299
*b**	1.361	0.634	1.578

a. Dependent variable: TPC.

## Data Availability

The datasets generated for this study are available on request to the corresponding author.

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
