# Peer review of "A Comprehensive Quality Analysis of Different Colors of Medicinal and Edible Honeysuckle"

_foods, 2023, doi:10.3390/foods12163126_

Round 1

Reviewer 1 Report

Honeysuckles is one of medicinal and edible substance that has great popularity among consumers due to its health effect. The price of the honeysuckle product was depends on the color of the product since it has different chemical compounds. Therefore, the exploration to compare the chemical compound based on honeysuckles' color is needed. The author's study would be excellent for further utilization of honeysuckles as a medicinal and natural food source. However, it requires several improvements before it can be considered for publication.

Abstract:

- What is the therapeutic and health effects from honeysuckles? mention birefly

- Please rearrange the sentence since it is not effective, ambiguous, and did not use proper punctuation-"In order to comprehensive compare the quality of different color, 55 batches of honeysuckle samples were collected,

color parameters, chlorophylls content (chl), total phenol content (TPC), total flavonoid content (TFC), antioxidant activity (AA), main active compounds, and metabolites were analyzed.

- Add information why the honeysuckle product differ in many color?

- Please rearrange the sentence since it is not effective sentence - "The correlation results showed that a* and h* values were significantly correlated with all quality indicators and 10 main compounds and that color difference could directly reflect quality and clinical efficacy."

Keywords:

- Please use different words from the manuscript title to enhance the discoverability

Introduction

- Line 29: Did author mean Introduction? 

- Line 42: What kind of beverage? since tea is a group of beverage too

- why does the color of honeysuckles can influence the quality and the price?

- add information why the honeysuckle product differ in many color?

Materials and Methods

- Please write detailed reagent that used in this study at the reagent section

- Temperature unit should be written separately from the value 4 °C

Result and Discussion

- Significant figure should be used the same decimal number

- Why this study chose rutin not quercetin as the standard of Total Flavonoid Content?

- Why this study chose to evaluate antioxidant activity with three different assay?

- What was the conclusion from the value of three different assay?

- At which wavelength the HPLC analyses was conducted?

- Why color is important indicator of food and TCMs?

Conclusion

- Add detailed information why GW consider as the best product of honeysuckle also

- Put completely all the result in conclusion also in abstract

Overall

- Please check again the used of punctuation and unit.

Author Response

Thanks for your suggestion, these responses are as follows:

Abstract:

Point 1: What is the therapeutic and health effects from honeysuckles? mention birefly.

Response 1: The the therapeutic and health effects from honeysuckles is brief mentioned by example in line 10.

Point 2: Please rearrange the sentence since it is not effective, ambiguous, and did not use proper punctuation-"In order to comprehensive compare the quality of different color, 55 batches of honeysuckle samples were collected, color parameters, chlorophylls content (chl), total phenol content (TPC), total flavonoid content (TFC), antioxidant activity (AA), main active compounds, and metabolites were analyzed.

Response 2: The above sentence in line 12 is divided into two sentences. ”In order to comprehensive compare the quality of different color, 55 batches of honeysuckle samples were collected and analyzed. Their color parameters, chlorophylls content (chl), total phenol content (TPC), total flavonoid content (TFC), antioxidant activity (AA), main active compounds, and metabolites were measured.”

Point 3: Add information why the honeysuckle product differ in many color?.

Response 3: The reason why honeysuckle products have many colors is described and added. It was merged with another sentence in line 10.

Point 4: Please rearrange the sentence since it is not effective sentence - "The correlation results showed that a* and h* values were significantly correlated with all quality indicators and 10 main compounds and that color difference could directly reflect quality and clinical efficacy."

Response 4: The above sentence in line 24 is rearranged. “The correlation results showed that a* and h* values were significantly affect above quality indicators and 10 main active compounds. In other words, color difference could directly reflect quality and clinical efficacy.” 

Keywords:

Point 5: Please use different words from the manuscript title to enhance the discoverability.

Response 5: The keywords in line 29 are revised to enhance the discoverability.

Introduction

Point 6: Line 29: Did author mean Introduction?

Response 6: Yes, the “Instruction” has been revised to “Introduction”.

Point 7: Line 42: What kind of beverage? since tea is a group of beverage too.

Response 7: The beverage is wine, the “beverages” has been revised to “wines” in line 44.

Point 8: why does the color of honeysuckles can influence the quality and the price?

Response 8: Because previous studies have shown that the greener the honeysuckle, the higher the content of phenolic acids such as chlorogenic acid and isochlorogenic acid A. This description and the corresponding references have been added to line 50 and reference 19, respectively.

Point 9: add information why the honeysuckle product differ in many color?

Response 9: The reason why honeysuckle products have many colors is described and added to line 55.

Materials and Methods

Point 10: Please write detailed reagent that used in this study at the reagent section.

Response 10: The detailed reagents that used in this study at the reagent section are added to line 77-92.

Point 11: Temperature unit should be written separately from the value 4 °C.

Response 11: Temperature unit in this manuscript has been revised and written separately from the value xx °C.

Result and Discussion

Point 12: Significant figure should be used the same decimal number.

Response 12: The significant figure has been revised to the same decimal number..

Point 13: Why this study chose rutin not quercetin as the standard of Total Flavonoid Content?

Response 13: This rutin is chosed as the standard of total flavonoid content is due to the determination method specified in the Chinese Pharmacopoeia and the previous literatures.

  • China, T.P.C.O. Pharmacopoeia of the People's Republic of China (Part 1). China Medical Science and Technology Press: Beijing, 2020; pp. 230-232, ISBN 978-7-5214-1574-2.
  • Liu, C.; Zhang, Z.; Dang, Z.; Xu, J.; Ren, X. New insights on phenolic compound metabolism in pomegranate fruit during storage. Hortic.2021, 285, 110138.
  • Li, C.X.; Zhao, X.H.; Zuo, W.F.; Zhang, T.L.; Zhang, Z.Y.; Chen, X.S. Phytochemical profiles, antioxidant, and antiproliferative activities of red-fleshed apple as affected by in vitro              digestion. Food Sci. 2020, 85, 2952-2959.

Point 14: Why this study chose to evaluate antioxidant activity with three different assay?

Response 14: Antioxidant activity is evaluated based on the free radical scavenging capacity, oxidation and reduction capacity of the sample in the detection system under specific conditions. In this study, the comprehensive evaluation of antioxidant activity by three different methods can make the experimental results more objective and authentic. In addition, there are a lot of reports using three and more methods to determine the antioxidant activity.

  • Yilmaz-Ersan, L.; Ozcan, T.; Akpinar-Bayizit, A.; Sahin, S. Comparison of antioxidant capacity of cow and ewe milk kefirs. Dairy Sci. 2018, 101, 3788-3798.
  • Li, C.X.; Zhao, X.H.; Zuo, W.F.; Zhang, T.L.; Zhang, Z.Y.; Chen, X.S. Phytochemical profiles, antioxidant, and antiproliferative activities of four red‐fleshed apple varieties in China. Food Sci. 2020, 85, 718-726.
  • Elisha, I.L.; Dzoyem, J.; McGaw, L.J.; Botha, F.S.; Eloff, J.N. The anti-arthritic, anti-inflammatory, antioxidant activity and relationships with total phenolics and total flavonoids of nine South African plants used traditionally to treat arthritis. BMC Altern. Med.2016, 16.
  • Cao, W.; Chen, J.; Li, L.; Ren, G.; Duan, X.; Zhou, Q.; Zhang, M.; Gao, D.; Zhang, S.; Liu, X. Cookies fortified with Lonicera japonica extracts: impact on phenolic acid content, antioxidant activity and physical properties. Molecules.2022, 27, 5033.
  • Amrani-Allalou, H.; Boulekbache-Makhlouf, L.; Mapelli-Brahm, P.; Sait, S.; Tenore, G.C.; Benmeziane, A.; Kadri, N.; Madani, K.; Jesus, M.M.A. Antioxidant activity, carotenoids, chlorophylls and mineral composition from leaves of Pallenis spinosa: An Algerian medicinal plant. J Complement Integr Med.2019, 17.

Point 15: What was the conclusion from the value of three different assay?

Response 15: The conclusion from the value of three different assay was improved and added in line 406-410. In details, these results showed that the antioxidant activity of honeysuckle gradually decreases with the deepening of color. Although antioxidant activity was lower in YW samples than in GW samples, the decrease was weak. It further suggested that YW samples might not suffer from the same conditions as YG, Y, and YB samples.

Point 16: At which wavelength the HPLC analyses was conducted?

Response 16: The wavelength is 325 nm. This parameter has been added to line 226.

Point 17: Why color is important indicator of food and TCMs?

Response 17: The color attribute is a important indicator of food and TCMs, which can reflect many characteristics such as environment conditions, pH value, and others. The United States Department of Agriculture (USDA) uses color to determine the safety and health of many foods, as does China. At present, there are many reports about importance and influence of color on the quality of foods and TCMs.

  • Kubec, R.; Urajová, P.; Lacina, O.; Hajšlová, J.; Kuzma, M.; Zápal, J. Allium discoloration: Color compounds formed during pinking of onion and leek. Agric. Food. Chem.2015, 63, 10192-10199.
  • Wibowo, S.; Vervoort, L.; Tomic, J.; Santiago, J.S.; Lemmens, L.; Panozzo, A.; Grauwet, T.; Hendrickx, M.; Van Loey, A. Colour and carotenoid changes of pasteurised orange juice during storage. Food Chem.2015, 171, 330-340.
  • Xia, Y.; Chen, W.; Xiang, W.; Wang, D.; Xue, B.; Liu, X.; Xing, L.; Wu, D.; Wang, S.; Guo, Q.; et al. Integrated metabolic profiling and transcriptome analysis of pigment accumulation in Lonicera japonicaflower petals during colour-transition. BMC Plant Biol. 2021, 21.
  • Yu, K.; Zhou, H.; Zhu, K.; Guo, X.; Peng, W. Physicochemical changes in the discoloration of dried green tea noodles caused by polyphenol oxidase from wheat flour. LWT. 2020, 130, 109614.

Conclusion

Point 18: Add detailed information why GW consider as the best product of honeysuckle also put completely all the result in conclusion also in abstract.

Response 18: The detailed information why GW consider as the best product of honeysuckle is added to line 643.

Overall

Point 19: Please check again the used of punctuation and unit.

Response 19: All punctuations and units are checked again.

Best wishes

Yours

Reviewer 2 Report

The presented paper is well-organized and well-written, focusing on the analysis of the correlation between color coordinates of honeysuckles and their quality parameters.  The authors collected 55 batches of honeysuckle samples for the study, and the research is well-planned, utilizing appropriate and modern methods.  The results are adequately described and thoroughly discussed.  The authors discovered that color coordinates were significantly correlated with the quality indicators and main compounds of honeysuckles, making this study highly practical.  Additionally, they examined the metabolic profile of honeysuckle. However, before publication, I have a few comments that should be taken into consideration.

Major revisions:

Lines 61-68: The information included in the Materials and Methods section should be removed.  Instead, please include the aim of the work.

Particle size of pulverized fractions of honeysuckles: Please provide the information on the particle size of the pulverized fractions of honeysuckles in the Materials and Methods section.

Conclusion: Please delete the last sentence and include the results of the correlation analysis.

Conclusion: The statement "As the color darkens or lightens, the quality gradually decreased" needs further clarification.  Please explain more precisely. Does it mean that lighter color is better or not?

Conclusion: Please expand the conclusion to highlight the most important and strongest correlations between color coordinates and quality parameters or chemical compound content.

Why the authors did not perform multivariate regression analysis? With fifty-five honeysuckle samples, there are enough data points for such an analysis. In this case, quality parameters can be expressed as a function of a few color coordinates.

Minor revisions:

Fig. 1: Please explain all abbreviations used under the figures or tables for better clarity.

The quality of some figures, for example, 2-4, 9-11, is poor.  Please consider improving the quality of these figures to enhance readability and understanding.

Minor editing of English language is required.

Author Response

Thanks for your review and suggestion, this response is as follows:

Major revisions:

Point 1: Lines 61-68: The information included in the Materials and Methods section should be removed. Instead, please include the aim of the work.

Response 1: The information has been removed, the aiming and content of this work have been added and improved.

Point 2: Particle size of pulverized fractions of honeysuckles: Please provide the information on the particle size of the pulverized fractions of honeysuckles in the Materials and Methods section.

Response 2: The information on the particle size of the pulverized fractions of honeysuckles is provided in line 98.

Point 3: Conclusion: Please delete the last sentence and include the results of the correlation analysis.

Response 3: The last sentence is deleted and the results of the correlation analysis is added.

Point 4: Conclusion: The statement "As the color darkens or lightens, the quality gradually decreased" needs further clarification. Please explain more precisely. Does it mean that lighter color is better or not?

Response 4: The word is revised to “After the color change, the quality parameters of honeysuckle decreased to different degrees”, which leaves no doubt about the statement.

Point 5: Conclusion: Please expand the conclusion to highlight the most important and strongest correlations between color coordinates and quality parameters or chemical compound content.

Response 5: The multivariate regression analysis is carried out and added in conclusion to highlight the importance between color coordinates and quality parameters.

Point 6: Why the authors did not perform multivariate regression analysis? With fifty-five honeysuckle samples, there are enough data points for such an analysis. In this case, quality parameters can be expressed as a function of a few color coordinates.

Response 6: In fact, we have already completed this part of the work before the manuscript is submitted. We have added this work to result section according to your suggestion. Thanks again for your extremely valuable suggestion.

Minor revisions:

Point 7: Fig. 1: Please explain all abbreviations used under the figures or tables for better clarity.

Response 7: All abbreviations used under the figures or tables are explained for better clarity.

Point 8: The quality of some figures, for example, 2-4, 9-11, is poor.  Please consider improving the quality of these figures to enhance readability and understanding.

Response 8: The quality of all figures in the manuscript is improved to enhance readability and understanding.

Point 9: 9.Comments on the Quality of English Language. Minor editing of English language is required.

Response 9: The English language is checked and revised again.

Best wishes

Yours

Round 2

Reviewer 1 Report

The author has revised and accommodated reviewers' suggestions clearly and completely. Therefore this manuscript can be considered for publication.

Reviewer 2 Report

Tha authors corrected the manuscript accordingly. However, the quality of figures is still very poor.